# Comorbidity, Radiation Duration, and Pretreatment Body Muscle Mass Predict Early Treatment Failure in Taiwanese Patients with Locally Advanced Oral Cavity Squamous Cell Carcinoma after Completion of Adjuvant Concurrent Chemoradiotherapy

**DOI:** 10.3390/diagnostics11071203

**Published:** 2021-07-02

**Authors:** Yu-Ching Lin, Hang Huong Ling, Pei-Hung Chang, Yi-Ping Pan, Cheng-Hsu Wang, Wen-Chi Chou, Fang-Ping Chen, Kun-Yun Yeh

**Affiliations:** 1Department of Medical Imaging and Intervention, Chang Gung Memorial Hospital, College of Medicine, Keelung & Chang Gung University, Taoyuan 333007, Taiwan; yuching1221@gmail.com; 2Osteoporosis Prevention and Treatment Center, Chang Gung Memorial Hospital, Keelung 20401, Taiwan; fangping@cgmh.org.tw; 3Division of Hemato-Oncology, Department of Internal Medicine, Chang Gung Memorial Hospital, College of Medicine, Keelung & Chang Gung University, Taoyuan 333007, Taiwan; xianfang87@gmail.com (H.H.L.); ph555chang@gmail.com (P.-H.C.); chw0098@gmail.com (C.-H.W.); 4Department of Nutrition, Chang Gung Memorial Hospital, Keelung 20401, Taiwan; pyngpyng@gmail.com; 5Division of Hemato-Oncology, Department of Internal Medicine, Chang Gung Memorial Hospital, College of Medicine, Linkou & Chang Gung University, Taoyuan 333007, Taiwan; wenchi3992@yahoo.com.tw; 6Department of Obstetrics and Gynecology, Chang Gung Memorial Hospital, Keelung 20401, Taiwan; 7Healthy Aging Research Center, College of Medicine, Chang Gung University, Taoyuan 333007, Taiwan

**Keywords:** oral cavity cancer, concurrent chemoradiotherapy, early treatment failure, body muscle mass, DXA

## Abstract

Few prospective cohort trials have evaluated the potential risk factors of early treatment failure of locally advanced oral cavity squamous cell carcinoma (LAOCSCC) patients following the completion of postoperative adjuvant concurrent chemoradiotherapy (CCRT). We collected clinicopathological variables, nutrition-inflammatory markers and total body composition data assessed by dual-energy X-ray absorptiometry (DXA) before and after CCRT. A factor analysis was used to reduce the number of DXA-derived parameters. Cox proportional hazard models were applied to determine the risk factors associated with early treatment failure defined as tumor progression or death within 180 days of CCRT completion. A total of 69 patients were eligible for analysis. After CCRT, the body weight, body mass index, nutritional markers, and muscle mass decreased, whereas C-reactive protein level increased. Five factors reflecting different body composition statuses were identified. A total of 21 patients (30.4%) developed early treatment failure. Comorbidities (hazard ratio ((HR)), 2.699; 95% confidence interval ((CI)), 1.005–7.913; *p* = 0.044), radiation duration (HR, 1.092; 95% CI, 1.015–1.174; *p* = 0.018) and the pretreatment body muscle mass (HR, 0.578; 95% CI, 0.345–0.957; *p* = 0.037) independently contributed to early treatment failure. Comorbidities, longer radiation duration, and lower pretreatment body muscle mass are predictive factors for early treatment failure in LAOCSCC patients following postoperative adjuvant CCRT completion.

## 1. Introduction

Concurrent chemoradiotherapy (CCRT) is the main treatment modality for patients with locally advanced head and neck cancer (LAHNC) following surgery as adjuvant therapy. Although its efficacy improves treatment outcomes and survival [1], a significant proportion of patients still experience treatment failure defined as tumor progression [2,3], and deaths from cancer or non-cancer causes such as toxicity and severe infection [4,5]. Particularly, early treatment failure, which occurs within a short period after CCRT completion, is a serious complication that may occur despite patients receiving adequate supportive care during CCRT [6,7]. Since the majority of patients with early tumor progression are asymptomatic, unaware of symptoms, or fail to bring it forward during their medical visits due to comorbidities, poor performance status, or fear of previous CCRT-associated adversities, a delayed detection of early treatment failure is commonly seen in the daily practice [8]. This delay may have profound implications for further treatment and long-term survival [8]. Thus, the prompt identification of the potential risk factors associated with ETF is urgently needed.

Certain clinicopathologic variables and nutritional-inflammatory markers (NIMs) correlated with early treatment failure in LAHNC patients following CCRT completion have been reported [4,7,9]. The risk factors include old age [4], poor performance status [4,7], comorbidities [4,9], low body mass index (BMI) and body weight (BW) [4,6,7], anemia [4,9], and a low total lymphocyte count [7]. However, the application of these risk factors to clinical practice requires further consideration. First, enrollment heterogeneity in most studies with a retrospective design could not be avoided. These retrospective reports analyzed the pretreatment clinicopathological conditions and NIMs [4,7,9]; however, the reports rarely addressed the role of treatment-interval change in the nutritional status. Second, some studies prospectively recruited LAHNC patients with identical tumor stages and used standard treatment protocols; however, their restrictive inclusion criteria excluded severe comorbidities, chronic consumption of alcohol, and tobacco use. Although they reported early toxic death events with associated etiology, the related prognostic outcomes of these subsets of patients are nonexistent. In addition, endpoints of these prospective studies did not observe tumor progression events shortly after treatment completion or the effect of total body composition on the prognosis [3,10,11,12,13,14,15,16,17]. Lastly, blood NIMs are interrelated and their influences are usually confounded by comorbidities or treatment-related toxicity rather than malnutrition alone [6]. Treatment-related toxicity exerts a detrimental effect on disease control and remains one of the major causes of unplanned schedule interruption, prolonging the overall treatment duration [18]. To identify the potential risk factors of early treatment failure, a prospective study recruiting LAHNC patients with homogeneous tumor location, stage, histology and treatment modality, and simultaneously analyzing the interplay effects between clinicopathological characteristics, treatment-related variables (dose and toxicity), and nutritional profile should be conducted.

Dual-energy X-ray absorptiometry (DXA) is a non-invasive clinical imaging method that has become the gold standard for the evaluation of lean mass, fat mass, and bone mineral content of the total body because it can precisely quantify each parameter with low radiation and costs [19]. Changes in the total body composition are common among cancer patients in response to aging, metabolic demand, physiological alterations, and therapy. Accordingly, monitoring the fluctuation in the total body composition, instead of the changes in the BW, BMI, or blood NIMs, may provide a more precise assessment of the nutritional/inflammatory changes during treatment [20]. Accumulating evidence has shown that DXA successfully assessed the TBC of CCRT-treated LAHNC patients [21,22,23]. At the time of diagnosis, LAHNC patients had lower lean body mass (LBM) and total fat mass (TFM) values than healthy adults [23]. Throughout the CCRT course, DXA detected continuous declines in the BW, LBM, and TFM [21,22,23,24]. Nonetheless, no study has examined the effects of DXA-derived parameters and their temporal changes on the early treatment failure in LAHNC patients.

In Taiwan, the incidence of oral cavity cancer is high, accounting for nearly 70% of newly diagnosed head and neck cancers [25]. It is associated with exposure to betel nut, a carcinogenic agent to humans. The male gender with an age range of 40~60 years is predominant in this illness [25]. Patients with resectable oral cavity cancer receive radical surgery and adjuvant radiotherapy or CCRT to improve disease control and survival. Oral cavity cancer is a multifactorial disease and patients are treated with multimodal therapy, but most studies searching for independent risk factors of treatment outcomes mixed head and neck cancer entities and were not exclusively limited to oral cavity cancer [26,27]. Consequently, clinicians may have difficulty in applying these data to patients with oral cavity cancer in daily practice.

Thus, we conducted a prospective observational cohort study and enrolled patients with locally advanced (stage III, IVA, or IVB) oral cavity squamous cell carcinoma (LAOCSCC) who received adjuvant CCRT following curative surgery. All patients had a supportive care program consisting of intensive symptom control, biweekly dietitian visits, and adequate daily calorie supplements at a single institution. In this study, we analyzed relevant information including clinicopathological variables, treatment-related factors, NIMs, and DXA-derived parameters to identify the potential risk factors responsible for early treatment failure in LAOCSCC patients following postoperative adjuvant CCRT completion.

## 2. Materials and Methods

This prospective cohort study was conducted between February 2015 and September 2018. This study was approved by the Institutional Review Board of the Chang Gung Memorial Hospital, Taiwan (approval numbers: 103-3365A3 and 201700158B0) and was performed in accordance with good clinical practice guidelines and the Declaration of Helsinki. All patients provided written informed consent prior to their inclusion in the study.

### 2.1. Enrollment

The eligible patients were aged 20–75 years with histologically proven LAOCSCC. The disease was classified according to the 7th edition of the American Joint Committee on Cancer staging system, which included stages III (T1-2, N1 or T3, N0-1), IVA (T4a, N0-1 or T1-4a, N2), and IVB (any T, N3 or T4b, any N). All the eligible patients had an Eastern Cooperative Oncology Group (ECOG) performance status score ≤2 with adequate hematopoietic or organ function and could undergo CCRT. The patients were excluded if they had end-stage renal failure, liver cirrhosis with intractable ascites, heart failure with New York Heart Association Classification IV, autoimmune disorders, major gastrointestinal disorders, uncontrolled diabetes mellitus, ongoing infections, or if they were receiving regular medications that could substantially modulate the metabolism or weight, such as steroids or megestrol acetate.

### 2.2. Treatment Schedule

Adjuvant CCRT was conducted in patients after surgery if they had (1) one of the two major risk factors of extranodal extension or a positive surgical margin; or (2) at least three of the following minor risk factors: pT4, pN1, close margin ≤ 4 mm, poor differentiation, perineural invasion, vascular invasion, lymph node invasion, or depth ≥ 10 mm. During CCRT, radiotherapy (RT) was delivered at a dose of 64–72 Gy in 32–36 fractions over a period of 6–8 weeks, and concurrent chemotherapy with cisplatin (40 mg/m^2^) was administered once weekly.

All patients received antiemetic medications and were routinely referred to an early and intensive nutritional support program established in 2007 in our institute that included biweekly dietitian visits, mandatory feeding tube placement if the BW loss was >5% during the treatment course, timely caloric supplementation, and blood transfusions as needed [28].

According to the guidelines of the European Society of Parenteral and Enteral Nutrition [28,29], energy requirement for each patient was estimated at 25–30 kcal/kg/day with energy percentage from carbohydrate: lipid = 60:40, and 1.0–1.5 g/kg/day of protein on the basis of nutritional status evaluated by patient-generated subjective global assessment (PG-SGA) during the CCRT course. Once patients could not attain the required daily calories via food, Isocal, an oral commercially available nutrition formula (Nestle, Taiwan, Ltd.; 1.06 kcal/mL, 250 kcal/237 mL; proteins, 17% of calories; lipids, 37% of calories; and carbohydrates, 46% of calories), was given. The purpose of this intensive nutrition support program was to enable each patient to achieve and maintain calculated energy and protein requirements during CCRT.

Approximately 95% of the patients were admitted to the hospital for completion of the treatment course and received government healthcare support via the Taiwan National Health Insurance Program.

### 2.3. Clinicopathological Data and NIMs

Clinicopathological data were collected, including age, sex, body height, BW, ECOG performance status, comorbid diseases, tumor location, tumor stage, treatment modality including the chemotherapy and radiotherapy doses, and history of smoking, alcohol consumption, and betel nut consumption. The severity of comorbid diseases was scored using the head and neck Charlson Comorbidity Index (HN-CCI), which was used to assess the presence of heart failure, pulmonary disease, cerebrovascular disease, peptic ulcers, liver disease, and diabetes [30]. Participants were considered smokers if they currently smoked cigarettes or had smoked in the past. Participants were considered alcohol drinkers if they reported consuming alcohol ≥ 4 times per week. Participants were considered betel nut users if they reported taking this substance during the previous year. BMI was defined as the weight in kilograms divided by the height in square meters (kg/m^2^). The PG-SGA scores ranged from 0 to 35, with scores of 0–3 indicating well-nourished, 4–8 indicating moderately malnourished, and ≥9 indicating severely malnourished [31]. The RT dose was defined as the radiation dose received by the patients during CCRT. RT duration was defined as the number of days that the patients took to complete RT. Cisplatin dose was defined as the cumulative dose of cisplatin administered during CCRT.

The blood NIMs before CCRT including hemoglobin (Hb, g/dL), white blood cell count (WBC, 10^3^/mm^3^), platelet count (10^3^/mm^3^), albumin (g/dL), and C-reactive protein (CRP, mg/dL) were examined.

### 2.4. Body Composition Measurement

The total body composition was measured using dual-energy fan-beam X-ray absorptiometry (Lunar iDXA, GE Medical System, Madison, WI, USA). The scan mode (standard, thin, or thick) was selected automatically by the scanner software according to body size and BMI. Scans were analyzed using enCORE Software, version 15 (GE Lunar, Chicago, IL, USA). Each participant was positioned according to the guidelines set by the International Society for Clinical Densitometry [32]. DXA was used to acquire the following parameters: LBM, TFM, appendicular skeletal mass index (ASMI), android, gynoid and bone mineral content (BMC). LBMI was calculated as LBM in kilograms divided by height in meters squared (kg/m^2^). TFMI was calculated as TFM in kilograms divided by height in meters squared (kg/m^2^). BMCI was calculated as BMC in kilograms divided by height in meters squared (kg/m^2^). All above parameters were analyzed. The interval changes (∆) of the above parameters throughout the CCRT course were also calculated and analyzed.

All blood NIMs and DXA studies were completed one week prior to initiating CCRT and within one week of CCRT completion.

### 2.5. Follow-Up and Early Treatment Failure

After treatment completion, the patients were followed up twice during the first month and then once a month. Each follow-up consisted of complete physical examination, a review of the patients’ current symptoms, and head and neck area check-up with fiberoptic endoscopy by HNC surgeons. Chest X-ray film and head and neck imaging using computerized tomography scan or magnetic resonance imaging were arranged at the third and sixth month of the scheduled follow-up or at a time of clinically suspicious tumor progression. According to the diagnosis recommendations formulated by the head and neck cancer committee in our institute, the patients who had tumor progression detected by computerized tomography scan or magnetic resonance imaging were required to complete positron emission tomography for further confirmation. Biopsy for histologically verified malignant tumor cells of the same type as the primary tumor site was arranged if image report and clinical examination were discrepant. All data were periodically reviewed by the head and neck cancer committee of our institute.

The early treatment failure rate was defined as the proportion of patients who had tumor progression or succumbed to cancer and non-cancer etiology within 180 days of CCRT completion, which was used as the reference date due to variation from the time for stage workups.

### 2.6. Statistical Analysis

The statistical analyses were performed using SPSS version 22.0 (SPSS Inc., Chicago, IL, USA). Based on the statistical analysis with a power of 80%, α error of 0.05, and LAOCSCC incidence rate in Taiwan, the minimum sample size was calculated to be 70. The patients with oral cavity cancer might not complete the CCRT course or the required data collection because of intolerance to treatment, poor compliance to medical advice, and insufficient family support. Subsequently, we estimated the attrition rate at 20%, increasing the total number of patients that needed to be recruited to 84. The descriptive statistics for all the variables, both continuous and categorical, were assessed for normality using the Kolmogorov–Smirnov normality test before analysis. Non-parametric paired tests for CRP and DXA-derived body composition parameters, and paired *t*-test for BMI, body weight, Hb, WBC, platelet count and albumin were used to detect the differences before and after treatment. The associations between the categorical variables were examined using Pearson’s chi-square test, and the continuous variables were compared using independent *t* test or nonparametric statistics with the Mann–Whitney test where appropriate.

Given that 12 variables were collected and calculated from DXA measurements, and high correlations existed among these variables, it was appropriate to reduce their number to minimize the loss of information. Thus, we applied the principal axis factor with a varimax (orthogonal) rotation of these 12 variables to conduct data derived from DXA. The Kaiser–Meyer–Olkin (KMO) measure of sampling adequacy was performed, and the minimum acceptable value of KMO was 0.6, although the idea was more than 0.7. Only factors with an eigenvalue of ≥1 were considered. The variance percentage accounted for by each component to the total variance was also reported. The factor score coefficient matrix was also computed.

The associations between different clinicopathological characteristics, treatment-related variables, toxicity profile, NIM, and DXA component parameters and early treatment failure rate were analyzed using Cox proportional hazard models. A forward stepwise selection was used in the univariate and multivariate analyses for different variables. All the independent variables significantly associated with an early treatment failure rate (*p* ≤ 0.05) in the univariate analysis were included in the multivariate analysis. The variance inflation factors were used to test for collinearity.

Receiver operating characteristic (ROC) curves were used to determine the optimal cutoff value when continuous variables showed significance in univariate analysis. The probabilities of early treatment failure-free survival were calculated using the Kaplan–Meier estimator. The log-rank test was used for comparing groups. All the differences in the early treatment failure rate were considered statistically significant with a *p*-value < 0.05 (two-tailed).

## 3. Results

### 3.1. Patient Characteristics

A total of 86 patients with locally advanced OCSCC were recruited, 69 of whom were eligible for analysis in this study. The patient enrollment, allocation, treatment modality, and data collection details are presented in a CONSORT diagram (Figure 1). The baseline patient characteristics are summarized in Table 1. All the patients were men and the mean age was 53.2 years. The most common tumor site was the tongue (40.6%), followed by the buccal mucosa (29.0%), and gingiva (18.8%). A high proportion of patients reported smoking (91.3%), alcohol consumption (73.9%), and betel nut consumption (78.8%). The majority of tumors were non-metastatic stage IV (94.2%), had an advanced tumor size (T4a and T4b, 75.2%), and had extensive regional lymph node involvement (N2 and N3, 66.5%). Forty patients (58.0%) had at least one comorbid illness and 46 (66.6%) underwent tracheostomy before CCRT. Regarding treatment conditions, more than 60% of patients received CCRT due to positive surgical margin or extrandoal extension. The mean radiation and cisplatin doses were 64.3 Gy and 238.5 mg/m^2^, respectively. The mean radiation duration was 48.0 days. For CCRT-related toxicity profile of grade 3 or higher, the most common non-hematologic adverse effects were mucositis (23.2%) and infection (14.5%), while the hematologic counterpart was neutropenia (33.3%). The mean daily calorie intake for each patient throughout the treatment course was 28.6 kcal/kg/day.

### 3.2. Anthropometric Data, NIMs, and DXA-Derived Parameters Before and after CCRT

Before treatment, the mean BMI was 22.8 ± 4.3 kg/m^2^, mean BW was 63.6 ± 12.6 kg, and 50% of patients were in a PG-SGA-defined severely malnourished status. At the end of the CCRT, the mean BMI was 21.8 ± 3.9 kg/m^2^ (mean 4.3% decrease from pretreatment, *p* < 0.001), the mean BW was 60.7 ± 11.2 kg (mean 4.6% decrease from pretreatment, *p* < 0.001), and more than 80% of patients were in a PG-SGA-defined severely malnourished status (Figure 2).

All blood NIMs showed significant changes at the end of treatment, except albumin (3.9 ± 0.5 vs. 3.8 ± 0.4 g/dL, *p* = 0.259). The nutrition-oriented markers including Hb, WBC, and platelet count were significantly decreased (Hb, 11.7 ± 1.5 vs. 10.6 ± 1.4 g/dL, *p* < 0.001; WBC, 7.3 ± 2.5 vs. 5.3 ± 2.7 10^3^ cells/mm^2^, *p* < 0.001; platelet count, 341.1 ± 148.4 vs. 245.7 ± 10.9 10^3^/mm^3^, *p* < 0.001). In contrast, there was a significant increase in the inflammation marker CRP (9.3 ± 13.7 vs. 18.2 ± 24.7 mg/dL, *p* = 0.037).

For DXA analysis, the muscle- and bone-related body composition parameters were significantly reduced after the completion of CCRT: LBMI (15.7 ± 1.6 vs. 14.8 ± 1.6 kg/m^2^, *p* < 0.001), ASMI (6.6 ± 1.0 vs. 6.0 ± 0.9 kg/m^2^, *p* < 0.001) and BMCI (1.37 ± 0.05 vs. 1.34 ± 0.04 kg/m^2^, *p* = 0.005). On average, patients lost 5.7% of their LBMI, 10.0% of ASMI, and 2.2% of BMCI over the course of CCRT. On the other hand, the TFMI (6.1 ± 3.1 vs. 5.8 ± 2.9 kg/m^2^, *p* = 0.064) and android percentage (29.6 ± 13.4 vs. 28.9 ± 13.1 %, *p* = 0.231) were not changed; however, the gynoid percentage (25.7 ± 8.3 vs. 28.8 ± 8.7 %, *p* = 0.006) was significantly increased after the completion of CCRT (Figure 2).

Despite certain changes in the body composition parameters at the end of CCRT, the total body composition ratio remained unchanged over the course of treatment. The mean total body composition was 69.2% LBMI, 29.1% ASMI, 26.9% TFMI, and 3.9% BMCI at the start of CCRT, respectively. At the end of treatment, the total body compositions were almost identical, with 68.8% LBMI, 27.9% ASMI, 26.9% TFMI, and 4.3% BMCI.

### 3.3. Factor Analysis

A principal axis factor analysis with varimax rotation was conducted to evaluate the underlying structure in all 12 DXA-derived variables that were conventionally classified into three categories. The body muscle category included LBMI, ASMI, ∆LBMI% and ∆ASMI%. The body fat category included TFMI, android, gynoid, ∆TFMI%, ∆android% and ∆gynoid%; the body bone category included BMCI and ∆BMCI%. The analysis yielded a five-factor solution; the KMO measure of sampling adequacy was 0.694, which is close to the ideal value. Table 2 shows the rotated component matrix, eigenvalues, and the percentage of variance explained; it is notable that the five components explain 89.2% of the variance.

Three variables (TFMI, android, and gynoid) loaded in Factor 1 accounted for 33.7% of the variance; these were all related to the body fat mass before CCRT. Three variables (∆TFMI%, ∆android%, and ∆gynoid%) loaded in Factor 2 accounted for 18.6% of the variance; these were all related to the interval change of the total body fat change throughout the CCRT course. Two variables (LBMI and ASMI) loaded in Factor, accounted for 16.6% of the variance; these two variables were related to the body muscle storage before CCRT. Two variables (∆LBMI% and ∆ASMI%) loaded in Factor 4 accounted for 11.6% of the variance; these two variables were related to the interval change of the total body muscle change throughout the CCRT course. Finally, one variable (BMCI) loaded in Factor 5 accounted for 8.7% of the variance; it represented the total body bone content (Table 2).

DXA, dual-energy X-ray absorptiometry; LBMI, lean body mass index; TFMI, total fat mass index; ASMI, appendicular skeletal muscle index; BMCI, bone mineral content index. Δ indicates a value obtained by subtracting the pretreatment value from the posttreatment value. % indicates (Δ value/the pretreatment value) × 100%.

### 3.4. Comorbidity, Radiation Duration, and Pretreatment DXA-Derived Muscle Mass Predict Early Treatment Failure

Overall, 21 patients (30.4%) experienced early treatment failure within 180 days of CCRT completion (Appendix A). Fifteen patients who developed tumor progression remained alive, four died of cancer, and two died of non-cancer etiology due to sepsis. The mean age was 53.1 years (range, 36–67 years). The mean time to develop early treatment failure was 3.5 months (range, 0.76–5.97 months) (Table 3). The patients with early treatment failure had more comorbidities (*p* = 0.042), poorer performance status (*p* = 0.018), higher CRP level (*p* = 0.017), lower Factor 3 expression derived from DXA before CCRT (*p* = 0.021) and required a longer time to complete the radiation course (*p* = 0.026) than those with no early treatment failure (Table 3).

We further examined the interactive effect among the clinicopathological variables, treatment dose and duration, treatment-induced toxicity, NIMs, and total body composition on early treatment failure. On univariate analysis, the following variables showed significance: comorbidities (HN-CCI), RT duration, grade 3/4 dermatitis during treatment, pretreatment CRP, and Factor 3. On multivariate analysis, comorbidities (*p* = 0.044), RT duration (*p* = 0.018), and pretreatment Factor 3 (*p* = 0.037) were the independent factors for early treatment failure following CCRT (Table 4).

Further analysis found that the patients with comorbidities, longer RT duration (RT days > 46 days, based on the ROC, area under curve [AUC]: 0.720; *p* = 0.004), or Factor 3 less than 0.67 (cutoff value based on the ROC, AUC: 0.704; *p* = 0.037) had higher early treatment failure rates (Figure 3).

## 4. Discussion

Three previous articles (one retrospective and two prospective designs) discussed the early treatment failure of LAHNC patients receiving postoperative or curative CCRT (Table 5). Although some risk factors related to early treatment failure have been reported, their application is limited because these studies analyzed the oral cavity and non-oral cavity cancers together, and the two CCRTs had different irradiation fields. The current study recruited a subgroup of LAHNC patients with homogenous stage and histology LAOCSCC who were receiving the standard adjuvant CCRT protocol, adequate supportive care, and nutrition monitoring, and we characterized changes in BW, BMI, NIMs, and total body composition assessed using DXA before and after treatment. At the end of the CCRT, the entire cohort experienced a reduction in BW, BMI, LBMI, ASMI, and BMCI with a varied extent but an increase in the gynoid percentage, which is comparable to previous reports [21,22,24]; the levels of nutrition-oriented NIMs, such as Hb, WBC, and the platelet count decreased. However, the level of CRP increased, similar to Moon‘s study, which prospectively analyzed 153 patients with non-oral cavity cancer receiving CCRT [33]. Importantly, we found that comorbidities, RT duration, and DXA-derived Factor 3 (body muscle mass before CCRT) were significantly associated with early treatment failure after controlling clinicopathological variables and NIMs. To the best of our knowledge, this study is also the first to demonstrate the effect of total body composition investigated by DXA on the prognosis of LAOCSCC patients treated with postoperative adjuvant CCRT.

Early treatment failure is defined as tumor progression or death within 180 days after CCRT completion and is applied as the endpoint based on the following evidence: Wong et al. found that 26.2% of the patients developed tumor progression within 6 months in 377 recurrent head and neck cancer patients who underwent primary curative surgery with or without adjuvant radiotherapy [34]; Kissun et al. detected early tumor recurrence as early as 3 months after treatment completion in patients with oral cavity and oropharyngeal cancer [8]. A large-scale retrospective study conducted in Taiwan analyzed 4839 recurrent head and neck patients without distant metastasis and found that 28.8% of the disease recurrence occurred within 3–6 months after treatment and patients with shorter recurrence-free intervals had a worse long-term survival [35]. Although routine follow-up schedules do not benefit the survival outcomes of LAHNC patients after treatment [36,37], an early awareness of the risk factors for early treatment failure in certain subgroups of LAHNC patients could improve the tumor control if salvage therapy is immediately provided. The current study identified three potential risk factors—comorbidities, RT duration, and body muscle mass before CCRT—to predict LAOCSCC patients possibly developing early treatment failure. To avoid a delay in the early treatment failure diagnosis, more attention should be paid to the patients with one of these three risk factors after treatment completion.

The risk factors associated with early treatment failure presented in this study were in partial accordance with previous reports that examined the survival outcomes [4,38,39,40,41]. A population-based study retrospectively analyzed the effect of comorbidity assessed using the Charlson Comorbidity Index (CCI) in 12,623 head and neck cancer patients and found that higher CCI scores had an inferior overall survival [38]. Qin et al. reported that comorbidities significantly increased the 90-day mortality and overall survival in 55,080 Taiwanese LAHNC patients after curative surgery [40]. The patients with pharyngeal and laryngeal cancers also had a high risk of early death after radical radiotherapy in the presence of multiple comorbidities [4]. Additionally, a prolonged radiation duration may accelerate the repopulation of surviving tumor cells and consequently correlates with poor treatment outcomes [42,43]. In the United States, one report analyzing 19,531 head and neck patients found that prolonged radiation duration was associated with worse OS in the patients who underwent primary RT, but had minimal effect on the overall survival in patients with postoperative RT [41]. Another study showed that prolonged radiation duration led to inferior overall survival in 129,055 head and neck patients who received either primary or postoperative RT [44]. A population-based study from the Taiwan Cancer Registry analyzed 8988 all-stage oral cavity squamous cell carcinoma patients who underwent curative surgery and adjuvant RT, and found that patients with RT duration over 8 weeks had worse overall survival, cancer-specific survival, and locoregional control [39]. The present study also showed significant association between the RT duration and early treatment failure rate in LAOCSCC patients who underwent curative surgery and postoperative adjuvant CCRT. Taken together, these reports including ours indicate that comorbidities and prolonged radiation duration are unfavorable prognostic factors for the short-term and long-term survival outcomes in head and neck cancer patients.

Since intimate and intricate correlations exist among the DXA-derived parameters, we categorized all the parameters into five factors, which indicated the body muscle-, fat- and bone-related mass before CCRT, and their temporal changes during CCRT, respectively. Only Factor 3, representing the body mass composed of LBMI and ASMI, was associated with early treatment failure after adjustment for all the covariates in this study. Neither fat- nor bone-related factors showed association with early treatment failure. Growing evidence has shown that head and neck cancer patients with low LBMI receiving RT or CCRT presented with higher treatment-related toxicity that consequently resulted in an early treatment termination, increased nosocomial infection and even mortality [45,46]. ASMI represents the physical activity and function of humans [47]. Cancer patients with low ASMI are known to have more bedridden experiences and often have complications such as infection and treatment-related toxicity; consequently, these toxicities may hamper the entire treatment course as planned [48]. Our analysis echoes these observations and found that the patients with low Factor 3 levels (≤0.67) had higher CRP values (*p* = 0.037) and higher incidences of Grade 3/4 toxicities including infection (*p* = 0.043), and neutropenia (*p* = 0.012) and required longer RT days (*p* = 0.031) than those with high Factor 3 level (>0.67) (Appendix A). Finally, the body muscle mass is able to generate energy via mitochondrial ATP synthesis [49], and its pretreatment status may represent the ability to offer extra energy for CCRT-treated head and neck cancer patients, who usually require more than 55,000 total calories to complete the treatment course [50]. From our own perspective, a good status of the pretreatment body muscle mass could help patients to tolerate CCRT-related toxicity and work as a sufficient energy reservoir to cope with the energy deficit during CCRT. Accordingly, the patients with sufficient pretreatment body muscle mass may have a better chance to complete the CCRT course and this may reduce the possibility of treatment failure or cancer progression.

Some study limitations merit further discussion. Because all patients were men and enrolled from the Taiwanese population, the corresponding sex, ethnicity, and the regional vulnerability to this disease remain unclear; hence, our results should be cautiously extrapolated to non-Taiwanese patients, or patients undergoing different CCRT schedules and nutrition support programs. Furthermore, head and neck cancer patients may continue to endure ongoing nutrition impact symptoms, which often result in an inadequate dietary intake and BW loss lasting for several months after CCRT completion. Our previous report found that a low BMI at 3 months after adjuvant CCRT completion affects the survival outcomes of head and neck cancer patients [51]. Whether the posttreatment nutrition status affects the early treatment failure rate remains to be investigated further. Additionally, this study followed the European Society of Parenteral and Enteral Nutrition guidelines and offered an average 28.6 ± 8.6 kcal/kg/day for each patient during CCRT. However, we observed that the BMI, BW, LBMI, ASMI, and the levels of blood nutrition-oriented markers (Hb, WBC, and platelet count) were decreased, while that of the inflammation-orient marker (CRP) was increased at the end of CCRT. Thus, these results suggest that a conventional protein-energy supply may not counteract increasing CCRT-driven inflammation, which induces cachexia-associated symptoms manifested by BW loss, nutrition-oriented marker decline, and muscle mass decrease. Nutritional intervention via supplementation with anti-inflammatory food components such as omega-3 fatty acids, selenium, and probiotics may attenuate inflammation generation and maintain the BW during treatment [52,53]. The impact of CCRT on myelosuppression that leads to declines in the Hb level, WBC, and platelet counts should also be considered in this study. Finally, 16 patients (18.6%) could not be analyzed due to an incomplete CCRT course or missing data collection during treatment. The aim of this study was to assess the potential risk factors pertaining to early treatment failure of LAOCSCC patients who were able to complete adjuvant CCRT, so those who failed to complete the course and comply with the data collection schedule were not eligible for final analysis. We observed that eight patients with an incomplete CCRT course developed unexpected sepsis, cardiovascular accident or severe pneumonia, resulting in the discontinuation of the treatment course. Among eight patients with incomplete data collection, five patients had a delayed scheduled DXA examination, and three patients missed the blood tests. The incomplete CCRT/data collection group had more comorbidities (*p* = 0.049) and exposure to betel nut exposure (*p* = 0.032), lower platelet count (*p* = 0.003), and a higher TFMI value (*p* = 0.001) than the complete CCRT group (Appendix A). Although these factors could affect the CCRT completion, the selection bias remains inevitable in this study because of the small sample size and analysis of two groups (incomplete CCRT and missing data collection). A prospective large-scale study examining the factors pertaining to the capability of LAOCSCC patients to complete postoperative adjuvant CCRT following curative surgery is warranted.

## 5. Conclusions

The current prospective observational study facilitates the clinical prediction of early treatment failure among patients with LAOCSCC after postoperative adjuvant CCRT completion using clinicopathological variables (comorbidities), treatment condition (RT duration), and nutritional status (body muscle mass before CCRT assessed by DXA).

## Figures and Tables

**Figure 1 diagnostics-11-01203-f001:**
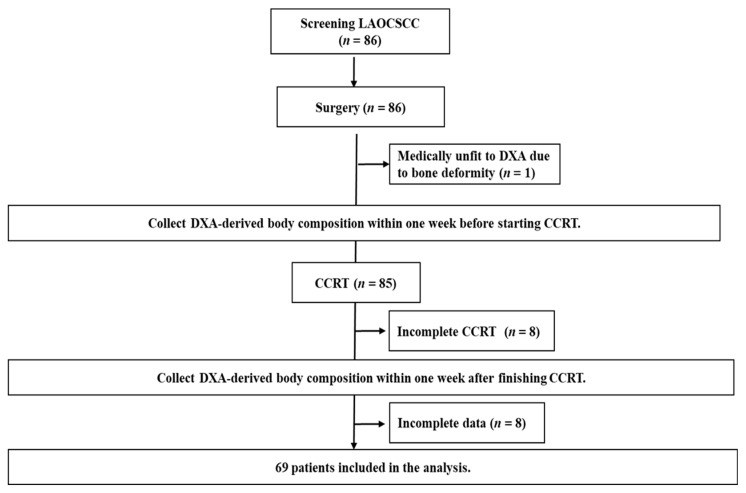
CONSORT diagram. CCRT was considered incomplete when patients dropped out during the treatment or failed to receive at least 4 cycles of weekly cisplatin (40 mg/m^2^) concomitant with planned radiotherapy (64–72 Gy). Incomplete data indicates that patients did not complete the required DXA examinations or missed the scheduled blood tests. LAOCSCC, locally advanced oral cavity squamous cell carcinoma; DXA, dual-energy X-ray absorptiometry; CCRT, concurrent chemoradiotherapy.

**Figure 2 diagnostics-11-01203-f002:**
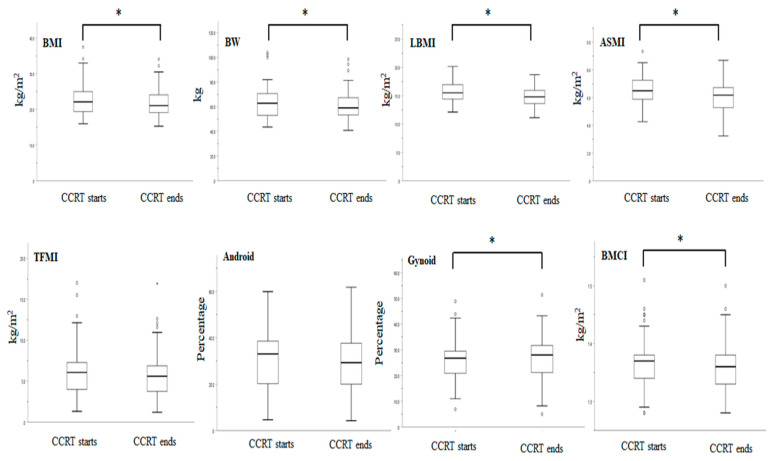
Box plots show the values of BMI, BW, and body composition parameters at CCRT start and end. * denotes *p* < 0.05, considered significant between start and end. BMI, body mass index; BW, body weight; LBMI, lean body mass index; ASMI, appendicular skeletal mass index; TFMI, total fat mass index; BMCI, bone mineral content index; CCRT, concurrent chemoradiotherapy.

**Figure 3 diagnostics-11-01203-f003:**
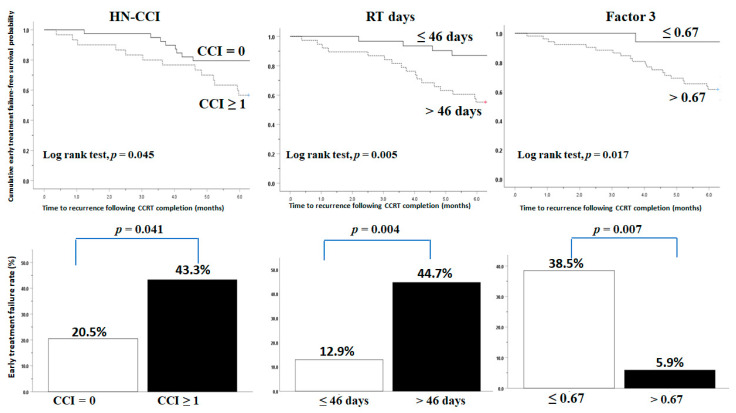
Early treatment failure-free survival probability or early treatment failure rates stratified by comorbidity, RT days (radiation duration), and Factor 3 (body muscle mass before CCRT). HN-CCI, head and neck Charlson Comorbidity Index; RT, radiotherapy.

**Table 1 diagnostics-11-01203-t001:** Baseline and treatment characteristics of 69 oral cavity cancer patients following postoperative CCRT completion.

Variables	Numbers (%) or Mean ± SD
Included patient number	69 (100.0)
Age (years)	53.2 ± 8.4
Sex (male: female)	69 (100.0):0 (0.0)
Tumor location	
Buccal mucosa	20 (29.0)
Tongue	28 (40.6)
Gingiva	13 (18.8)
Mouth floor	3 (4.3)
Retromolar	2 (2.9)
Lip	2 (2.9)
Hard palate	1 (1.4)
TNM stage (III:IVA:IVB)	4 (5.8):50 (72.5):15 (21.7)
Tumor size (T1:T2:T3:T4a: T4b)	2 (2.9):6 (8.7):11 (15.9): 45 (65.3):5 (7.2)
Lymph node involvement (N0:N1:N2:N3)	21 (30.5):9 (13.0):29 (42.0):10 (14.5)
Histological grade (1:2:3)	8 (11.6):51 (73.9):10 (14.5)
Smoking (no:yes)	6 (8.7):63 (91.3)
Alcohol (no:yes)	18 (26.1):51 (73.9)
Betel nut (no:yes)	16 (23.2):53 (76.8)
HN-CCI (0:1:2:≥3)	29 (42.0):15 (21.7):6 (8.7):19 (27.5)
ECOG performance status (0:1:2)	2 (2.9):61 (88.4):6 (8.6)
Tracheostomy (no:yes)	23 (33.4):46 (66.6)
Adjuvant CCRT due to risk factor	
One major (positive surgical margin or ENE)	42 (60.9)
≥3 Minors	27 (39.1)
Adjuvant CCRT	
Radiotherapy	
Dose (Gy)	64.3 ± 3.8
Fractions	32.0 ± 1.6
Duration (days)	48.0 ± 4.9
Cisplatin dose (mg/m^2^)	238.5 ± 20.5
PG-SGA (well:moderate:severe)	2 (2.9):32 (46.4):35 (50.7)
Anthropometric and biochemical data before CCRT	
BW (kg)	63.6 ± 12.6
BWL (%)	1.0 ± 6.3
BMI (kg/m^2^)	22.8 ± 4.3
<18.5:≥18.5	14 (20.3):55 (79.7)
Hb (g/dL, normal range: 12.0–16.0)	11.7 ± 1.5
WBC (×10^3^ cells/mm^3^, normal range: 6.0–11.0)	7.3 ± 2.5
Platelet count (×10^3^/mm^3^, normal range: 150–450)	341.1 ± 148.4
Albumin (g/dL, normal range: 3.5–4.5)	3.9 ± 0.5
CRP (mg/dL, normal range: <5.0)	9.3 ± 13.7
DXA-related parameters	
LBMI (kg/m^2^) before CCRT	15.7 ± 1.6
TFMI (kg/m^2^) before CCRT	6.1 ± 3.1
ASMI (kg/m^2^) before CCRT	6.6 ± 1.0
Android (%) before CCRT	29.7 ± 13.4
Gynoid (%) before CCRT	25.7 ± 8.3
BMCI (kg/m^2^) before CCRT	1.4 ± 0.1
∆LBMI% *	−5.68 ± 0.72
∆TFMI% *	−2.22 ± 1.97
∆ASMI% *	−5.48 ± 0.52
∆Android% *	0.34 ± 2.59
∆Gynoid% *	4.37 ± 1.76
∆BMCI% *	−0.23 ± 0.12
Mean daily calorie intake during CCRT (kcal/kg/day)	28.6 ± 8.6
CCRT	
Radiotherapy	
Dose (Gy)	64.3 ± 3.8
Fractions	32.0 ± 1.6
Duration (days)	49.6 ± 6.6
Cisplatin dose (mg/m^2^)	238.5 ± 45.5
Toxicity during CCRT	
Non-hematologic (any grade:grade 3/4)	
Dermatitis	62 (89.9):3 (4.3)
Pharyngitis	24 (34.8):5 (7.2)
Infection	13 (18.8):10 (14.5)
Mucositis	27 (39.1):16 (23.2)
Emesis	33 (47.8):6 (8.7)
Hematologic (any grade:grade 3/4)	
Anemia	(95.7):5 (7.2)
Neutropenia	57 (82.6):23 (33.3)
Thrombocytopenia	42 (60.9):4 (5.8)
Early treatment failure (%)	21(30.4)
Tumor progression (%)	15 (21.7)
Death by cancer (%)	4 (5.8)
Death by non-cancer (%)	2 (2.9)

Abbreviations: CCRT, concurrent chemoradiotherapy; SD, standard deviation; SCC, squamous cell carcinoma; HN-CCI, head and neck cancer-Charlson Comorbidity Index; ECOG, Eastern Cooperative Oncology Group; patient-generated subjective global assessment; BW, body weight; BWL, body weight loss; BMI, body mass index; Hb, hemoglobin; WBC, white blood cell; CRP, C-reactive protein; BUN, blood urea nitrogen; ALT, alanine aminotransferase; DXA, dual-energy X-ray absorptiometry; LBMI, lean body mass index; TFMI, total fat mass index; ASMI, appendicular skeletal muscle index; BMCI, bone mineral content index. * Δindicates a value obtained by subtracting the pretreatment value from the posttreatment value. % indicates (Δvalue/the pretreatment value) × 100%.

**Table 2 diagnostics-11-01203-t002:** Factor analysis results of DXA-related parameters among 69 patients following postoperative CCRT completion.

Factors					
Component	1	2	3	4	5
**LBMI**	0.214	−0.120	0.947	−0.038	0.104
**ASMI**	0.226	−0.126	0.943	−0.108	−0.024
**TFMI**	0.926	−0.053	0.313	−0.031	−0.008
**Android**	0.931	−0.076	0.207	−0.071	−0.120
**Gynoid**	0.970	−0.064	0.031	−0.054	0.011
**BMCI**	−0.015	−0.062	0.045	0.094	0.885
**∆LBMI%**	−0.055	0.011	−0.129	0.931	−0.096
**∆ASMI%**	−0.085	0.125	−0.006	0.940	−0.001
**∆TFMI%**	−0.095	0.901	−0.206	0.283	−0.031
**∆Android%**	−0.153	0.905	−0.104	0.088	0.136
**∆Gynoid%**	0.016	0.918	0.036	−0.173	0.024
**∆BMCI%**	0.089	−0.248	−0.025	0.340	−0.645
**Eigenvalue**	4.04	2.24	1.99	1.39	1.04
**% of accumulative variances**	33.7	52.3	68.9	80.5	89.2

**Table 3 diagnostics-11-01203-t003:** The characteristic variations between patients with and without early treatment failure in 69 oral cavity cancer patients following CCRT completion.

Early Treatment Failure			
Variables, Expressed as Numbers (%) or Mean ± SD	No	Yes	*p*-Value
***Patient number***	48	21	
***Clinicopathologic***			
Age (years)	53.3 ± 8.7	53.1 ± 8.1	0.954
Tumor location			0.298
Buccal mucosa	10 (20.8)	10 (47.6)	
Tongue	20 (41.6)	8 (38.1)	
Gingiva	11 (22.9)	2 (9.5)	
Mouth floor	2 (4.2)	1 (4.8)	
Retromolar	2 (4.2 )	0 (0.0)	
Lip	2 (4.2 )	0 (0.0)	
Hard palate	1 (2.1)	0 (0.0)	
TNM stage (III vs. IVA vs. IVB)	2 (4.2):35 (72.9):11 (22.9)	2 (9.5):15 (71.4):4 (19.1)	0.661
T status (T0-2 vs. T3-4)	7 (14.6):41 (85.4)	1 (4.8):20 (95.2)	0.241
N status (N0-1 vs. N2-3)	21 (43.8):27 (56.3)	9 (42.9):12 (57.1)	0.945
Histological grade (1:2:3)	2 (4.2):35 (72.9):11 (22.9)	2 (9.5):15 (71.4):4 (19.1)	0.065
ECOG performance status (0:1:2)	1 (2.1):46 (95.8):1 (2.1)	1 (4.8):15 (71.4):5 (23.8)	0.018 *
Smoking (no:yes)	4 (8.3):44 (91.7)	2 (9.5):19 (90.5)	0.872
Alcohol (no:yes)	15 (31.3):33 (68.7)	3 (14.3):18 (85.7)	0.140
Betel nut (no:yes)	13 (27.1):35 (72.9)	3 (14.3):18 (85.7)	0.246
HN-CCI (0 vs. 1 vs. 2 vs. ≥3)	26 (54.2):10 (20.8):5 (10.5):7 (14.5)	7 (33.3):5 (23.8):1 (4.8):8 (38.1)	0.042 *
Tracheostomy (no:yes)	16 (33.3):32 (66.7)	7 (33.3):14 (66.7)	1.000
Adjuvant CCRT due to risk factor			0.514
One major (positive surgical margin or ENE)	28 (58.3)	14 (66.7)	
≥3 Minors	21 (41.7)	7 (33.3)	
***CCRT***			
RT dose (Gy)	64.0 ± 3.3	64.9 ± 4.8	0.382
RT duration (days)	47.2 ± 4.6	50.0 ± 5.0	0.026 *
Cisplatin dose (mg/m^2^)	237.8 ± 15.8	240.0 ± 28.9	0.687
***CCRT-induced grade 3/4 toxicity***			
Dermatitis	1 (2.1)	2 (9.5)	0.163
Pharyngitis	2 (4.2)	3 (14.3)	0.136
Mucositis	9 (18.8)	7 (33.3)	0.253
Infection	6(12.5)	4 (19.0)	0.477
Emesis	5 (10.4)	1 (4.8)	0.443
Anemia (%)	4 (8.3)	1 (4.8)	0.599
Neutropenia (%)	16 (33.3)	7 (33.3)	1.000
Thrombocytopenia (%)	4 (8.3)	0 (0.0)	0.173
***Mean daily calorie intake during CCRT (kcal/kg/day)***	28.1 ± 8.1	29.8 ± 9.9	0.439
***Nutritional and inflammatory markers before CCRT***			
BMI (kg/m^2^)	23.3 ± 4.5	21.4 ± 3.3	0.090
BWL (kg)	1.7 ± 22.3	4.5 ± 15.2	0.276
Hb (mg/dL)	11.9 ± 1.5	11.2 ± 1.4	0.082
WBC (×10^3^ cells/mm^3^)	7.1 ± 2.1	7.7 ± 3.4	0.372
Platelet count (×10^3^/mm^3^)	330.4 ± 150.6	365.3 ± 124.4	0.370
Albumin (g/dL)	3.9 ± 0.5	3.8 ± 0.5	0.526
CRP (mg/dL)	8.5 ± 10.4	17.9 ± 22.0	0.017 *
***PG-SGA (well* vs. *moderate vs. severe) before CCRT***	2 (4.2):23 (47.9):23 (47.9)	0 (0.0):9 (42.9):12 (57.1)	0.549
***Body composition parameters***			
Factor 1	0.037 ± 1.037	−0.085 ± 0.935	0.643
Factor 2	0.045 ± 1.030	−1.033 ± 0.943	0.574
Factor 3	0.181 ± 0.979	−0.415 ± 0.969	0.021 *
Factor 4	−0.089 ± 0.942	0.205 ± 1.116	0.263
Factor 5	−0.005 ± 0.885	0.012 ± 1.247	0.940
***Time to treatment failure (months)***	30.9 ± 13.4	3.5 ± 1.6	<0.001 *

* indicates a significant *p*-value < 0.05. Abbreviations: CCRT, concurrent chemoradiotherapy; TNM, tumor node metastasis; ECOG, Eastern Collaboration Oncology Group; HN-CCI, head and neck Charlson Comorbidity Index; RT, radiotherapy; PG-SGA, patient-generated subjective global assessment; BMI, body mass index; BWL, body weight loss; Hb, hemoglobin; WBC, white cell count; TLC, total lymphocyte count; CRP, C-reactive protein. Independent *t* test was used for age, BMI, Hb, platelet count, and albumin. Mann–Whitney test was used for BWL, WBC, CRP, and all body composition parameters.

**Table 4 diagnostics-11-01203-t004:** Univariate and multivariate Cox regression analyses of factors associated with early treatment failure rate of 69 oral cavity cancer patients following CCRT completion.

Variables	Early Treatment Failure		
	Univariate	Multivariate	
	*p*-Value	Hazard Ratio (95% Confidence Interval)	*p*-Value
***Clinicopathologic***			
Age	0.988		
TNM stage (ref: IV)	0.620		
T status (ref: T3-4)	0.285		
N status (ref: N2-3)	0.423		
ECOG performance status (ref: 2)	0.221		
Smoking (ref: yes)	0.845		
Alcohol (ref: yes)	0.176		
Betel nut (ref: yes)	0.300		
HN-CCI (ref: no)	0.047 *	2.699 (1.005–7.193)	0.044 *
Tracheostomy (ref: yes)	0.920		
Risk factor for CCRT (ref: minor)	0.505		
***CCRT***			
RT dose	0.254		
RT duration	0.035 *	1.092 (1.015–1.174)	0.018 *
Cisplatin dose	0.886		
***CCRT-induced grade 3/4 toxicity***			
Dermatitis (ref: yes)	0.030 *	0.236 (0.040–1.383)	0.180
Pharyngitis (ref: yes)	0.102		
Mucositis (ref: yes)	0.495		
Infection (ref: yes)	0.423		
Emesis (ref: yes)	0.418		
Anemia (ref: yes)	0.644		
Neutropenia (ref: yes)	0.990		
Thrombocytopenia (ref: yes)	0.416		
***Mean daily calorie intake during CCRT (kcal/kg/day)***	0.108		
***Nutritional and inflammatory markers before CCRT***			
BMI	0.094		
BWL	0.315		
Hb	0.069		
WBC	0.681		
Platelet count	0.408		
Albumin	0.433		
CRP	0.012 *	1.018 (0.995–1.041)	0.130
***PG-SGA (well* vs. *moderate vs. severe) before CCRT***	0.976		
***Body composition parameters***			
Factor 1	0.636		
Factor 2	0.570		
Factor 3	0.008 *	0.578 (0.345–0.957)	0.037 *
Factor 4	0.339		
Factor 5	0.772		

* Indicates a significant *p*-value < 0.05 Abbreviations: CCRT, concurrent chemoradiotherapy; TNM, tumor node metastasis; ECOG, Eastern Collaboration Oncology Group; HN-CCI, head and neck Charlson Comorbidity Index; RT, radiotherapy; PG-SGA, patient-generated subjective global assessment; BMI, body mass index; BWL, body weight loss; Hb, hemoglobin; WBC, white cell count; TLC, total lymphocyte count; CRP, C-reactive protein.

**Table 5 diagnostics-11-01203-t005:** Studies reporting early treatment failure in oral cavity cancer patients treated with adjuvant CCRT.

Study	Tumor Location	CCRT Type	Radiotherapy	Chemotherapy	Characteristics and Endpoint Assessment
Cooper et al., 2004 [14]	Oral cavity, oropharynx, hypopharynx, larynx	Adjuvant CCRT	60 Gy, 6 weeks	Cisplatin (triweekly, 100 mg/m^2^)	Prospective, 206 patients, early treatment failure rate (within 30 days after CCRT): 3.4% (7 patients, 4 died of treatment toxicity, 1 died of cancer), no risk factor reported
Bernier et al., 2004 [11]	Oral cavity, oropharynx, hypopharynx, larynx	Adjuvant CCRT	66 Gy, 5–6 weeks	Cisplatin (triweekly, 100 mg/m^2^)	Prospective, 167 patients, early mortality rate: 0.6% (one patient died of sepsis), no risk factor reported
Chang et al., 2013 [7]	Oral cavity, oropharynx, hypopharynx	Primary CCRT or adjuvant CCRT	60–74 Gy, 6–8 weeks	cisplatin alone, cisplatin with oral UFT (tegafur plus uracil and calcium folinate)	Retrospective, 194 patients, early mortality rate (within 60 days after CCRT): 7.2% (14 patients, 11 died of sepsis); risk factors: performance status > 1, BMI < 19 kg/m^2^, and blood TLC < 700/mm^3^
This study	Oral cavity	adjuvant CCRT	64–72 Gy, 6–8 weeks	Cisplatin (weekly, 40 mg/m^2^)	Prospective, 69 patients, early treatment failure rate (within 180 days after CCRT): 30.4% (21 patients, 15 with tumor progression, 4 died of cancer, 2 died of sepsis): risk factors: comorbidity, RT duration, and muscle mass before CCRT

CCRT, concurrent chemoradiotherapy; 5-FU, 5-fluorouracil; frx, fraction; BMI, body mass index; TLC, total lymphocyte count; Hb, hemoglobin.

## Data Availability

The data presented in this study are available on request from the corresponding author.

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
