# Peer review of "Comorbidity, Radiation Duration, and Pretreatment Body Muscle Mass Predict Early Treatment Failure in Taiwanese Patients with Locally Advanced Oral Cavity Squamous Cell Carcinoma after Completion of Adjuvant Concurrent Chemoradiotherapy"

_diagnostics, 2021, doi:10.3390/diagnostics11071203_

Round 1
Reviewer 1 Report
In the title of the paper the mistake “comorbodity” should be replaced by “comorbidity”.
Big number of abbreviations used all over the text, including abstract is cumbersome for potential reader.
In the Introduction section some more data concerning oral carcinoma, especially in Taiwanese population will be desirable.
Abbreviations used in Fig. should be explained below the figure 2 and 3.
In Table 3, the last variable i.e., “Time to early treatment failure” applied for non-early treatment failure group of patients (n=48) is unclear.
ROC curves used to determine the optimal cutoff value for some variables should be presented.
Reviewer 2 Report
The study aimed to search for new predictive marker (comorbidity, radiation duration, and pre-treatment body muscle mass) for patients with locally advanced oral cavity squamous cell carcinoma (OSCC) after completion of adjuvant concurrent chemotherapy. The manuscript is well written, and the methods seem to be adequate. My comments are as below:
Major Revisions
#1:
Please explain why the authors selected OSCC for the study. Because the treatment of OSCC leads to eating disorder? Because OSCC basically lead to poor survival rate?
#2: page 5 (2.6. Statistical Analysis):
I think “Kaiser-Meyer Olkin (KMO) measure of sampling adequacy” is not very popular method. Please clarify why the authors selected the method.
Minor Revisions
#3: Page 1, the Title:
Please correct “Comorbodity” to “Comorbidity”.
#4:
The manuscript has too many abbreviations to read. Please reduce.
For example, IRB, AJCC, ESPEN, CT, MRI, ROC, TNM might not be used (only using in the Tables or Figures is preferred).
#5: Page 3 (2.1. Enrollment):
Were the exclusion criteria according to the published article? Or, the original criteria of the authors? Please explain here.
#6: Page 7, Table 1:
Is the normal range really < 5.0 mg/dL? Or it might be 0.5 mg/dL? Please confirm.
#7: page 10, Table 2:
Please correct “BCMI” and “ΔBCMI%” to “BMCI” and “ΔBMCI%”, respectively.
#8: page 11, Table 3:
Please add “(months)” to the “Time to early treatment failure”.
Round 2
Reviewer 2 Report
Thanks for the revisions. The revisions are sufficient.